# Starch-Coated Magnetic Iron Oxide Nanoparticles for Affinity Purification of Recombinant Proteins

**DOI:** 10.3390/ijms23105410

**Published:** 2022-05-12

**Authors:** Vasilisa V. Krasitskaya, Alexander N. Kudryavtsev, Roman N. Yaroslavtsev, Dmitry A. Velikanov, Oleg A. Bayukov, Yulia V. Gerasimova, Sergey V. Stolyar, Ludmila A. Frank

**Affiliations:** 1Institute of Biophysics SB RAS, Federal Research Center “Krasnoyarsk Science Center SB RAS”, 660036 Krasnoyarsk, Russia; vasilisa.krasitskaya@gmail.com (V.V.K.); kirush07@mail.ru (A.N.K.); 2Kirensky Institute of Physics SB RAS, Federal Research Center “Krasnoyarsk Science Center SB RAS”, 660036 Krasnoyarsk, Russia; yar-man@bk.ru (R.N.Y.); dpona1@gmail.com (D.A.V.); helg@iph.krasn.ru (O.A.B.); jul@iph.krasn.ru (Y.V.G.); stol@iph.krasn.ru (S.V.S.); 3Federal Research Center “Krasnoyarsk Science Center SB RAS”, 660036 Krasnoyarsk, Russia; 4School of Fundamental Biology and Biotechnology, Siberian Federal University, 660041 Krasnoyarsk, Russia

**Keywords:** iron oxide nanoparticles, starch, affinity sorbent, maltose-binding protein, hybrid proteins purification

## Abstract

Starch-coated magnetic iron oxide nanoparticles have been synthesized by a simple, fast, and cost-effective co-precipitation method with cornstarch as a stabilizing agent. The structural and magnetic characteristics of the synthesized material have been studied by transmission electron microscopy, Mössbauer spectroscopy, and vibrating sample magnetometry. The nature of bonds between ferrihydrite nanoparticles and a starch shell has been examined by Fourier transform infrared spectroscopy. The data on the magnetic response of the prepared composite particles have been obtained by magnetic measurements. The determined magnetic characteristics make the synthesized material a good candidate for use in magnetic separation. Starch-coated magnetic iron oxide nanoparticles have been tested as an affinity sorbent for one-step purification of several recombinant proteins (cardiac troponin I, survivin, and melanoma inhibitory activity protein) bearing the maltose-binding protein as an auxiliary fragment. It has been shown that, due to the highly specific binding of this fragment to the starch shell, the target fusion protein is selectively immobilized on magnetic nanoparticles and eluted with the maltose solution. The excellent efficiency of column-free purification, high binding capacity of the sorbent (100–500 µg of a recombinant protein per milligram of starch-coated magnetic iron oxide nanoparticles), and reusability of the obtained material have been demonstrated.

## 1. Introduction

The number of studies on various biomedical and biotechnological applications of magnetic nanoparticles (MNPs) has been steadily increasing [1,2]. Due to the combination of the large specific surface area and unique magnetic properties, MNPs find application in cell separation [3], drug and gene delivery [4], magnetic resonance imaging (MRI) [5], minimally invasive surgery and hyperthermia [6], in vitro molecular diagnostics (biosensing) [7], and are used as affinity sorbents for isolation/pre-concentration of target molecules [8]. In separation and isolation of macromolecules, MNPs work similarly to sorbents of other types (latexes, polymer, and inorganic), but make it possible to use magnetic separation instead of centrifugation, filtration, or chromatography systems and thereby reduce the time of separation of target products and automate the process.

To be used as affinity sorbents, MNPs should meet certain requirements, specifically, to have desired magnetic properties, be resistant against microbes and enzymes, exhibit the “zero” adsorption capacity to prevent nonspecific interactions and the stability of the functional surface, and contain an immobilized ligand for forming specific complexes with target biomolecules. Among a wide range of MNPs, iron oxide ones were proven to be promising candidates for the role of sorbents, owing to the synthesis simplicity, high magnetization, and superparamagnetism [9]. For use as sorbents, colloidal solutions of MNPs are conventionally prepared. The behavior of nanoparticles in a magnetic field strongly depends on the stability of orientation of their magnetic moments against thermal fluctuations. Superparamagnetic nanoparticles are characterized by the hysteresis-free behavior and high colloidal stability. In the blocked state, nanoparticles exhibit the field hysteresis and high magnetic susceptibility, whereas in a colloid they are less stable. The point between the temperatures of the superparamagnetic and blocked states is called the blocking temperature (*T*_B_). This parameter of nanoparticles is important for biomedical applications, including magnetic separation. For use in magnetic separation, the blocking temperature of nanoparticles should be similar to room temperature.

The blocking temperature depends, in particular, on characteristic measurement time τ [10]. The blocking temperature and the characteristic measurement time are related by the Néel-Brown formula where *K* is the anisotropy constant, *V* is the particle volume, $\tau_{0}$ is the nanoparticle relaxation time, and $k_{B}$ is the Boltzmann constant.
(1)$$T_{B}=\frac{K⬝V}{k_{B}⬝\ln\left(\tau /\tau_{0}\right)},$$

The characteristic measuring time is ~10^2^ s for the quasi-static measurements, ~2.5 × 10^−8^ s for Mössbauer spectroscopy, and ~10^−10^ for X-band ferromagnetic resonance (FMR). Therefore, a method for determining the blocking temperature should be selected according to the field of application of nanoparticles. For example, the blocking temperature of nanoparticles for use in FMR hyperthermia [11,12] should be determined by the FMR method, while the blocking temperature of nanoparticles for magnetic separation [13,14] should be determined by magnetometry.

According to Equation (1), the blocking temperature depends also on the constant of the magnetic anisotropy, which involves three contributions: the magnetocrystalline, surface, and shape anisotropies. The blocking temperature of nanoparticles can be changed by changing their size and shape. The plate-shaped and cubic nanoparticles have a large anisotropy constant and, consequently, a high blocking temperature [15].

Since naked iron oxide MNPs are prone to oxidization and aggregation, different stabilizers and coating agents are used to enhance their stability. In addition, a coating can act as a spacer for attaching certain biomolecules to a magnetic carrier. The materials used for coating MNPs include inorganic compounds (silicon oxide, carbon, and noble metals) and synthetic (PEG, PVA, etc.) and natural (chitosan, polysaccharides, proteins, and peptides) polymers [9]. Different polysaccharides are also frequently used for this purpose due to their chemical and structural diversity, which provides an excellent opportunity to develop novel magnetic micro- or nanocomposites with the high sorption capacity. Various magnetic nanoparticles carrying cellulose, chitosan, arabinogalactan, dextran, and amylose were synthesized and then functionalized with specific biomolecules [16,17,18].

In this work, starch-coated iron oxide MNPs (starch-MNPs) were synthesized and tested as an affinity sorbent for purifying recombinant proteins genetically fused with maltose-binding protein (MBP), which is a common protein expression tag. This is a 42-kDa single-chain protein encoded by the *E. coli* malE gene [19], which specifically binds to maltose and amylopectin. MBP is widely used as a fusion partner in production of a recombinant protein in bacterial cells to increase the expression level, improve the folding and solubility of the protein of interest [20], and implement its one-step purification on amylose-activated resin. Purification is carried out under the mild elution conditions, which maintain the specific activity of a target protein [21,22,23]. MNPs were prepared from iron sulfate by a simple co-precipitation method with cornstarch (amylose-amylopectin copolymer) as a stabilizing agent. The method allowed us to synthesize starch-MNPs with the required magnetic response and dispersibility in aqueous solutions without additional surface functionalization and starch pretreatment stages.

The obtained starch-MNPs were tested as a sorbent for affine purification of several small and medium-sized MBP-fused recombinant proteins: cardiac troponin I (cTnI), a highly specific early biomarker for acute myocardial infarction [24] and survivin (Surv) and melanoma inhibitory activity protein (MIA), the cancer biomarkers of negative prognosis and decreased survival in cancer patients [25,26]. The synthesized starch-MNPs showed excellent purification efficiency, high binding capacity, stability, as well as reusability.

## 2. Results and Discussion

### 2.1. Physical Properties of Starch-MNPs 

The starch-MNPs were prepared by co-precipitation of iron sulfate and cornstarch (21–24% amylose) as a stabilizing agent.

Figure 1 shows high-resolution transmission electron microscopy (HRTEM) images, a microdiffraction pattern, and a size distribution of the starch-MNPs. The particles are cubic nanocrystals with an average size of ~11.5 nm. The microdiffraction pattern (Figure 1b) is characteristic of a spinel structure (magnetite or maghemite).

The Mössbauer spectrum (Figure 2) shows that all iron ions are in the trivalent state. The experimental spectrum fits well with the components shown. The results of the spectrum interpretation are given in Table 1. The fit error is 3%. Four sextets and one doublet were observed. The widths of 3–4 lines of the inner sextets are extraordinarily large, which is indicative of electron density fluctuations on iron nuclei (variations in chemical shifts on these sites). Therefore, the sample can be identified as extremely defective maghemite. Since only 11% of iron nuclei are characterized by a doublet, i.e., are in the superparamagnetic state, it can be concluded that the average blocking temperature is significantly higher than room temperature.

In studying polysaccharide-coated MNPs, Fourier-transform infrared (FTIR) spectroscopy is used, since the presence of a matrix does not prevent obtaining information about a particle and, in addition, makes it possible to establish changes in all main types of bonds between coating molecules.

The qualitative analysis of the spectra of starch and starch-MNPs allowed us to determine the main characteristic frequencies (Table 2). Figure 3 shows the FTIR spectra of the investigated samples. It can be seen that the spectra are different. It should be noted that the FTIR spectra of MNP-containing composites have a characteristic background increase in the region of 2000–4000 cm^−1^. In the spectrum presented in Figure 3b, bonds in the region of 1200–1500 cm^−1^ disappear, while bonds in the region of 800–1200 cm^−1^ and hydroxyl ones remain unchanged.

According to the data of the spectral analysis, the absorption peaks at 388 and 570 cm^−1^ in the spectrum of starch-MNPs correspond to O–Fe–O bending vibrations and Fe–O stretching modes and the starch chemisorption occurs on the nanoparticle surface through acetal bonds [27,28,29].

To determine the magnetic characteristics (magnetization, coercivity, and blocking temperature) of the synthesized nanoparticles, the hysteresis loops were measured in fields from −2 to 2 kOe at temperatures of 80–295 K. The saturation magnetization at room temperature was found to be 29.8 emu/g.

Figure 4a shows the hysteresis loops. The temperature dependence of the coercivity shown in Figure 4b obeys Equation (2), i.e., the coercivity decreases with an increase in temperature up to the blocking point [30]. This equation is conventionally used to describe single-domain noninteracting nanoparticles at temperatures below the blocking point [15,31,32]:(2)HcT=Hc0·1−TTBα.

Figure 4b shows, along with the temperature dependence of the coercivity, the fitting of the experimental data with Equation (2). The best fit parameters are $H_{c}\left(0\right)=173\;\mathrm{Oe}$, $T_{B}=377\;K$, and α=0.67.

The blocking temperature can also be estimated using Néel-Brown formula (1). Assuming the anisotropy constant to be 4 × 10^5^ erg/cm^3^ [33] and the particle size to be ~11.5 ± 3 nm (according to the TEM data), we obtain that the blocking temperature of a nanoparticle, depending on its size, is 70–390 K.

Thus, according to the data obtained by the two methods, the average blocking temperature of the starch-MNPs is similar to room temperature and therefore they can be used in magnetic separation.

### 2.2. The Use of Starch-MNPs for One-Step Affine Purification of the MBP-Fused Recombinant Proteins

The obtained starch-MNPs were tested as affine sorbents for purification of several recombinant hybrid proteins: MBP–cTnI, MBP–Surv, and MBP–MIA consisting of maltose-binding protein (MBP), cardiac troponin I (cTnI, 23.9 kDa), survivin (Surv, 16.5 kDa), and melanoma inhibitory activity protein (MIA, 12.1 kDa), respectively. As was mentioned above, MBP is widely used as an expression tag for producing recombinant proteins in bacterial cells. MBP as an auxiliary polypeptide increases the expression level of a target protein, improving the solubility and folding of the latter [20] and providing its one-step purification on amylose resin. Purification of the MBP-fusion protein exploits the natural affinity of MBP for α-(1–4) maltodextrin with the micromolar dissociation constants [34]. The protein domains were linked by the ENLYFQS peptide, a specific protease site of the tobacco etch virus (TEV protease site). With this protease, the auxiliary polypeptide can be selectively removed, if desired. The target proteins chosen in this work are of considerable interest for medical diagnostics. Cardiac troponin I is a highly specific early biomarker for acute myocardial infarction. The MIA protein was identified as a key molecule involved in progression and metastasis of malignant melanomas. Survivin known as a baculoviral inhibitor of apoptosis repeat-containing 5 (BIRC5) is a member of the inhibitor of apoptosis protein (IAP) family, which inhibits caspases and blocks cell death. It is highly expressed in cancer and associated with a poorer clinical outcome.

All the hybrid proteins were synthesized in the corresponding recombinant *E. coli* cells and, after ultrasonic disintegration, extracted from the cytoplasmic fraction. The proteins of interest were purified using a simple procedure, which does not require complex high-tech chromatographic equipment (see the detailed description in Section 3).

The purification process was controlled by sodium dodecyl sulfate–polyacrylamide gel electrophoresis (SDS-PAGE) (Figure 5). After washing off non-adsorbed proteins, the target proteins were efficiently eluted with 10 mM of maltose. It can be seen that, after one-step purification by starch-MNPs, the final protein samples had a purity of 80–94% (lanes 4 in Figure 5).

It was found that the total amount of the protein taken for chromatography and summed up from all the eluted fractions was almost the same. This points out the absence of protein loss caused by the nonspecific irreversible adsorption on starch-MNPs.

The binding capacities were found to be 100.5 ± 5.1 µg for MBP-cTnI, 177 ± 14.1 µg for MBP–MIA, and 587.2 ± 52.2 µg for MBP–Surv (per milligram of starch-MNPs). These values exceed the parameters of commercial amylose magnetic beads (the binding capacity of MBP5*-paromyosin ΔSal fusion protein is 10 µg per milligram of New England Biolabs amylose magnetic beads) and those of amylose-coated particles obtained by Lim et al. [35] (the binding capacity of GFP fused with MBP is 72 µg per milligram of particles).

Several experiments on changing the starch-MNP capacity were carried out. This parameter was found to be independent of the concentration of starch used in co-precipitation within 5–50 g/L (the data are not shown). In all the purification experiments, the nanoparticles prepared at a starch concentration of 20 g/L were used.

To evaluate the synthesis reproducibility, we obtained three starch-MNP samples in parallel experiments and tested them in the MBP–cTnI purification. The high-purity MBP-cTnI yields were found to be similar, with a variation coefficient of 14.2%.

The recycling property of starch-MNPs was evaluated by measuring the MBP-binding capacity after three cycles of their use. MBP-cTnI was purified for three times using the same starch-MNPs sample (see Section 3). The particles showed a remarkable reuse robustness: the purified MBP-cTnI yields were 94.7, 104.1, and 102.8 µg per milligram of starch-MNPs after the first, second, and third cycle, respectively. Thus, the MBP binding capacity of starch-MNPs remains unchanged after the three cycles of reuse. It is noteworthy that, in principle, due to the low cost of the synthesized particles, this material can be disposable.

The storage of starch-MNPs in distilled water without preservatives leads to the complete loss of the MBP binding capacity in three weeks. It was found, however, that the initial binding capacity can be kept for up to six months by adding 0.05% of NaN_3_.

## 3. Materials and Methods

### 3.1. Synthesis and Characterization of Nanoparticles

Iron oxide nanoparticles were prepared by co-precipitation from a solution containing 20 g/L of ammonium iron (II) sulfate, 0.2–0.4 M of sodium citrate, 20 g/L of EDTA-Na_2_, and 20 g/L of cornstarch. At a temperature of 80 °C, this solution was added with a 0.1 M of the sodium hydroxide solution until neutral pH. The starch-MNPs were thoroughly washed with distilled water to remove ions and stored with the 0.05% aqueous solution of NaN_3_.

The electron microscopy study was carried out on a Hitachi HT7700 transmission electron microscope at an accelerating voltage of 100 kV at the Krasnoyarsk Regional Center for Collective Use of the Krasnoyarsk Scientific Center, Siberian Branch of the Russian Academy of Sciences. Mössbauer spectra were recorded on an MS-1104Em spectrometer with a 57Co(Cr) source at room temperature. Isomer chemical shifts were referenced to α-Fe. IR absorption spectra were recorded on a Bruker VERTEX-80V vacuum FTIR spectrometer in the range of 380–7500 cm^−1^. The samples were pressed tablets with potassium bromide with a diameter of 13 mm and a thickness of ~0.55 mm. They were carefully ground with 0.2 g of KBr in a ratio of 2:100 in a mortar. The spectra were obtained using a Globar light source (a U-shaped silicon carbide arc), an RT-DLaTGS detector, and a KBr beam splitter. The experimental data were processed and analyzed using the OriginPro software (OriginPro 2015, OriginLab Corporation, Northampton, MA, USA). The static magnetic measurements were performed on an automated vibrating sample magnetometer in fields of up to 15 kOe at room temperature [36].

### 3.2. MBP Hybrid Protein Expression

The sequences of genes encoding human surviving, cardiac troponin I, and MIA optimized for bacterial expression (Evrogen, Moscow, Russia) [37], were cloned into the pMALc5x vector.

*E. coli* BL21-CodonPlus (DE3)-RIPL cells (Stratagene, La Jolla, CA, USA) transformed by the corresponding plasmid were cultivated in an LB medium containing 2 g/L of glucose and 200 μg/mL of ampicillin at 37 °C until the culture reached an OD_600_ of 0.5–0.7; then, 0.33 mM of IPTG was added and the cultivation was continued at 37 °C for 3 h. The cells were harvested by centrifugation, the pellet was resuspended in buffer A (20 mM Tris-HCl pH 7.5, 0.2 M NaCl, 1 mM EDTA) in a ratio of 1:5 (*w*/*v*), disrupted by sonication (20 s × 6) at 0 °C, and the mixture was centrifuged. The pellet was discarded and the supernatant was used in the further protein purification experiments.

### 3.3. MBP Hybrid Proteins Purification by the Magnetic Nanoparticles

The cell lysate supernatant (800 μL aliquots) was mixed with 500 μL of the starch-MNP suspension at 4 °C for 1 h under shaking. Then, the particles were fixed at the bottom with a magnet and the solution was removed. The particles were washed with 3 portions of buffer A (1 mL) and then the hybrid proteins were eluted with 800 μL of the elution buffer (10 mM of maltose in buffer A).

The purity of the protein samples was controlled by Laemmli electrophoresis. The protein concentration was measured using a Bio-Rad DCTM protein assay kit.

### 3.4. Starch-MNPs Recycling

Recycling of starch-MNPs was evaluated by quantifying the purified MBP-cTnI capacity for three additional loading cycles with 0.8 mL of cell lysate to 0.5 mL of the starch-MNP suspension. After the MBP-cTnI binding and elution (as described above), the starch-MNPs were regenerated by washing with the elution buffer for several times followed by re-equilibration with buffer A. Then, the cycle of regeneration, re-equilibration, binding, washing, and elution was repeated. These experiments were carried out for three times.

## 4. Conclusions

It was shown that starch-coated magnetic iron oxide nanoparticles synthesized by a simple and cost-effective co-precipitation method with cornstarch as a stabilizing agent represent a high binding affinity material that can ensure the high-efficiency purification of a recombinant protein bearing the maltose-binding protein as an auxiliary fragment. Starch-MNPs in the form of cubic nanocrystals with an average size of ~11.5 nm were synthesized. It was demonstrated that the saturation magnetization (29.8 emu/g at room temperature) and blocking temperature similar to room temperature (377 K, according to the temperature dependence of the coercivity, or 70–390 K, according to the Néel-Brown equation) make these nanoparticles good candidates for use in magnetic separation. The FTIR spectroscopy data disclosed the formation of a bond between a nanoparticle and starch.

Starch-MNPs exhibit the high specificity and purification capacity for MBP-fusion proteins (100–500 µg per milligram of the particles), as well as the reusability without the binding capacity and protein purity loss. They provide the fast, convenient, specific, and cost-effective one-step purification of recombinant proteins. The use of magnetic separation in the purification process makes it possible to exclude the centrifugation and filtration stages, as well as the need for expensive chromatographic systems and columns.

Thus, the obtained starch-MNPs have a high potential for use in purification of recombinant proteins. In our opinion, this material has also good prospects as a basis for the development of specific biosensors. The proposed simple and cost-effective starch-MNP synthesis method can be easily implemented at non-specialized (resource-limited) laboratories and facilities.

## Figures and Tables

**Figure 1 ijms-23-05410-f001:**
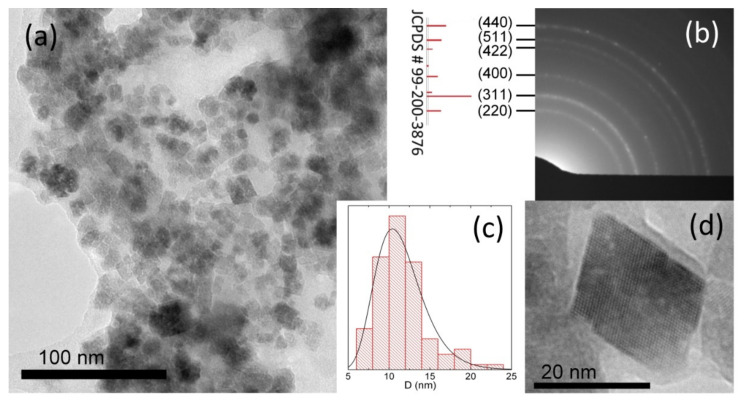
HRTEM images at (**a**) the ×100 k and (**d**) ×400 k magnifications, (**b**) microdiffraction pattern, and (**c**) size distribution of the synthesized iron oxide nanoparticles.

**Figure 2 ijms-23-05410-f002:**
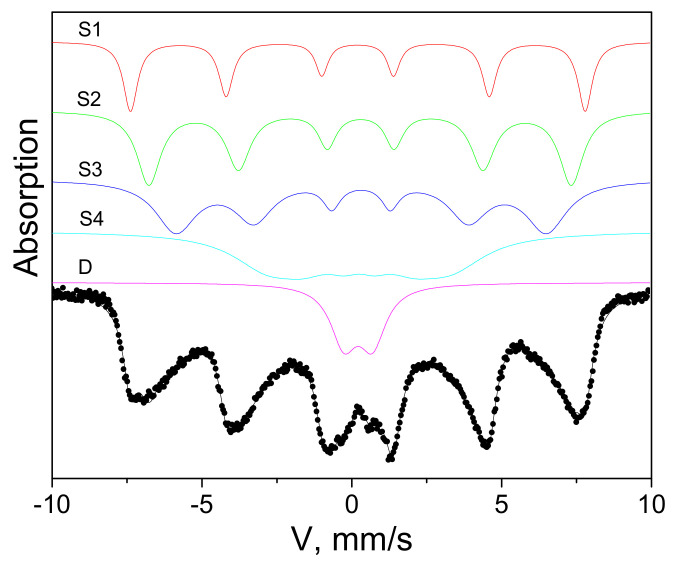
Mössbauer spectrum of starch-MNPs.

**Figure 3 ijms-23-05410-f003:**
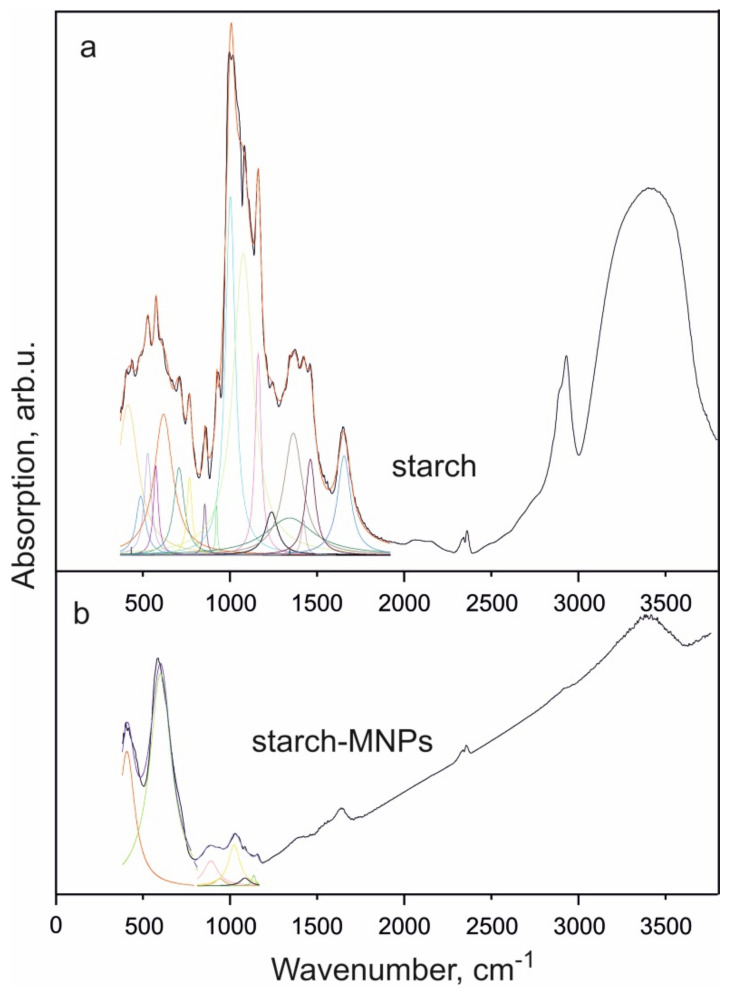
(**a**) FTIR spectrum of starch at *T* = 300 K and division of the spectral region at 380–1900 cm^−1^ into components. (**b**) FTIR spectrum of starch-MNPs and division of the spectral region at 380–1200 cm^−1^ into components.

**Figure 4 ijms-23-05410-f004:**
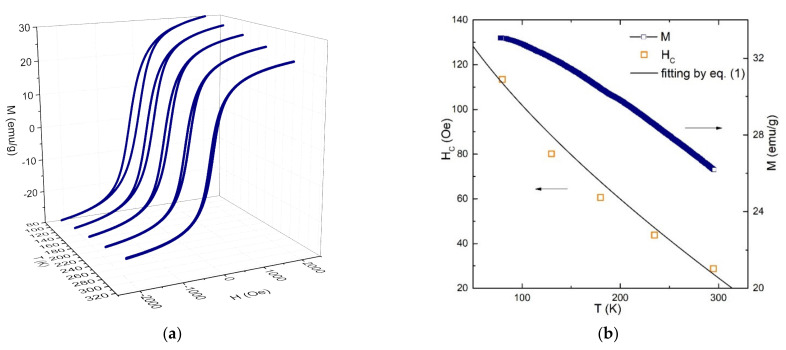
(**a**) Hysteresis loops measured in fields from –2 to 2 kOe in the temperature range of 80–295 K. (**b**) Temperature dependence of the coercivity (orange squares) fitted by Equation (2) (black line) and temperature dependence of magnetization measured in a field of 5 kOe (blue squares).

**Figure 5 ijms-23-05410-f005:**
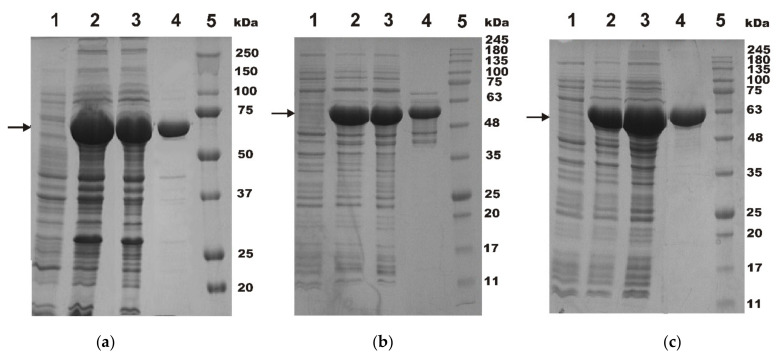
12.5% SDS-PAGE analysis of MBP-TnI (**a**), MBP-MIA (**b**) and MBP-Surv (**c**) purification using starch-MNPs. Lanes: 1—whole-cells lysates before IPTG induction; 2—whole-cell lysates after IPTG induction; 3—cytoplasmic fraction; 4—fractions after elution by 10 mM maltose; 5—standard proteins (BioRad, Hercules, CA, USA), molecular weights are shown with numbers. Arrows show hybrid proteins bands.

**Table 1 ijms-23-05410-t001:** Parameters of the Mössbauer spectrum of starch-MNPs.

IS, mm/s	H, kOe	QS, mm/s	W_34_, mm/s	A, Fract. %	
0.34	474	0.01	0.50	0.14	S1
0.42	440	−0.04	0.73	0.24	S2
0.45	386	0.01	0.67	0.26	S3
0.37	193	0.01	1.24	0.25	S4
0.35	-	0.96	1.08	0.11	D

IS is the isomer chemical shift, H is the hyperfine field on the iron nucleus, QS is the quadrupole splitting, W34 is the width of 3–4 lines of the inner sextets, and A is the fractional site population.

**Table 2 ijms-23-05410-t002:** Absorption peaks and their interpretation (cm^−1^).

Range	Absorption Peak, Starch	Absorption Peak, Starch-MNPs	Description
380–800		388	O–Fe–O
	413		Vibrations of the pyranose ring and δ-hydroxyl groups
	432		
	487		
	527		
	573		Vibrations of the chain C–C–C…–
		570	Fe–O
	617		Vibrations of the pyranose ring and δ-hydroxyl groups
	706		
	767		
800–1000	855		C–O in C–O–H
		866	
	922	900	
1000–1200	1001		C–O stretching of internal vibrations of C–O bonds (the bands characteristic of polysaccharides are caused by the presence of acetal bonds)
		1025	
	1075		
		1092	
		1150	
	1161		
1200–1500	1238		δ-CH_2_ groups in CH_2_OH
	1341		δ-O–H bonds in CH_2_OH
	1368		δ-bonds of CH_2_ groups
	1421		δ-CH_2_ groups
	1461		δ-OH
1500–2000	1654	1635	δ-bonds in H–O–H (adsorbed water)
2000–3000	2060		ν-bonds in CH and CH_2_ groups
	2153		
	2890		
	2930	2928	C–H
3000–4000	3406	3413	Internal vibrations of OH groups involved in intermolecular and intramolecular H bonds

## Data Availability

Not applicable.

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
