# Peer review of "Starch-Coated Magnetic Iron Oxide Nanoparticles for Affinity Purification of Recombinant Proteins"

_ijms, 2022, doi:10.3390/ijms23105410_

Round 1

Reviewer 1 Report

About this paper:  Starch-coated magnetic iron oxide nanoparticles for affinity pu- 2 rification of recombinant proteins. First of all, this paper is not specific enough. The English of this manuscript is extremely poor and parts of it are unintelligible. The authors must get it to read and corrected by a professional language service before considering resubmission to any journal.

 I think it can be not considered for submission.  I think this paper needs a major overhaul in order to make it more interesting and valuable for the readers of the journal.

1.    Please state more about salient findings in the abstract shortly.
2. Please use updated and recent papers in the literature review to give more sense to the reader.
3. Conclusions could be more specific and to the point, I would suggest looking and thinking about it.
4. Please more elaborate on the novel aspect of your work at the end of the introduction.
5. Please added  FTIR and FESEM for  NMP 
6. what is the  Reaction between Starch-coated magnetic iron oxide nanoparticles with protein?
7. Compare the traditional methods of analysis with the methods based on this subject. Maybe a table could help
8. Conclusion should be rewritten. Please added future perspective before the Conclusion
9. Do you mains about the economic process?

Author Response

Dear Editor,

On behalf of the authors, I thank the Reviewers for the comments, questions and remarks. We tried to take into account the comments and wishes of the reviewers. Corrections made to the text of the manuscript are shown in color.

Below are our responses to reviewer questions in order.

Reviewer 1

First of all, this paper is not specific enough. The English of this manuscript is extremely poor and parts of it are unintelligible. The authors must get it to read and corrected by a professional language service before considering resubmission to any journal.

                        ­ The English of this manuscript has been corrected

  1. Please state more about salient findings in the abstract shortly.

                        ­ The abstract has been edited

  1. Please use updated and recent papers in the literature review to give more sense to the reader.

                        ­ Several recent publications have been added in the literature review

  1. Conclusions could be more specific and to the point, I would suggest looking and thinking about it.

                        ­ Conclusions have been corrected to be more specific

  1. Please more elaborate on the novel aspect of your work at the end of the introduction.

                        ­ Introduction has been edited

  1. Please added FTIR and FESEM for NMP

                        ­ Unfortunately, we do not have the opportunity to add these measurements, but we have greatly expanded the discussion of infrared spectroscopy data. In addition, we believe that FESEM measurements will not provide new information in addition to TEM and Mössbauer spectroscopy.

  1. What is the reaction between Starch-coated magnetic iron oxide nanoparticles with protein?

                        ­ The target proteins have been fused genetically with maltose (maltodextrin)-binding protein (MBP). MBP is monomeric globular protein with molecular weights of 42 kDa, containing a single ligand-binding site for carbohydrate α-(1–4) maltodextrin (Kd∼3.5 μM). Cornstarch consist of amylose (25-75%) and amylopectin. Of all E.coli proteins, only MBP-target protein hybrid are able specifically bound the amylose on the starch-MNPs surface. All other proteins remain in solution and are discarded. The target hybrid protein are eluted by 10 mM maltose.

This is a known approach that is used among others to purify recombinant proteins. In our work, we obtained a new material on the base of MNP, with suitable magnetic properties and the ability to bind MBR-bearing proteins. We have shown the promise of its use for fast and efficient purification of target proteins bearing MBP without the use of expensive equipment.

Some explanations about this have been added to the manuscript.

  1. Compare the traditional methods of analysis with the methods based on this subject. Maybe a table could help

                 ­ It is not entirely clear what the reviewer means by traditional methods of analysis? Did you mean traditional protein purification methods? Affinity chromatography of MBP fusion proteins is usually performed on an FPLC system with an amylose column or batch-wise using amylose agarose resin, followed by elution by solution of 10 mM maltose. Due to the magnetic properties of the starch-MNPs synthesized in our work, it is possible to use magnetic separation and eliminate the use of centrifugation, filtration, or expensive chromatographic systems (and a packed column), significantly simplifying the process and reducing the purification time. Besides, as we noted in the mns, the genetic extending of the target protein with MBP improves its solubility and folding during E.coli sintheses. The use of specific linkers between the MBP and the target protein allows selective removal of the auxiliary fragment, if neсessary.

 Conclusion should be rewritten. Please added future perspective before the Conclusion

                        ­ Conclusions have been rewritten, future perspective have been added.

  1. Do you mains about the economic process?

                        ­ The availability commercial plasmid constructs, containing MBP gene for cytoplasmic expression of fused target protein and a range of suitable E.coli host cells (free of charge and have very reasonable licensing and royalty terms) make MBP based recombinant carrier protein expression one of the most economically achievable systems available for both research and commercial ventures. The obtained starch-MNPs provide a fast, convenient, specific and cost-effective one-step purification of the recombinant proteins. The magnetic separation based on starch-MNPs eliminates centrifugation, filtration and the use of expensive chromatographic systems and columns. In addition, method co-precipitation used for synthesizing of starch-MNPs is the most simple and cost-effective method known. The use of starch during the synthesis of MNPs as a stabilizing agent makes it possible to cover nanoparticles surface by amylose available for MBP binding without any additional stages of surface functionalization or pretreatments of starch. All of these simplifies and reduces the cost of affine magnetic material synthesis.

The authors hope that the changes made have significantly improved the manuscript and made it more understandable and useful for the reader.

On behalf of coauthors,

 Dr. Ludmila A. Frank.

Reviewer 2 Report

The work entitled
"Starch-coated magnetic iron oxide nanoparticles for affinity purification of recombinant proteins" 
by Vasilisa V. Krasitskaya
is a reasonable piece of work.

It brings only incremental contributions, with routine methods, to an intensely worked domain. However, given the relevance of the topic, it deserves publication, after certain improvements.

Please define, in causal manner, how the magnetic properties are working for the declared application purposes.
Also, make explicit the implied recombination mechanism. Whap about specificity?
Add naked MNPs reference to ftir spectrum. 

To define the Fe-O bonding some x-ray surface spectroscopy would be useful.
Explain how the dicrimination of bonded starch layer from supllemenary outer ones is possible. How tick is the starch coating? Can it be stripped?

Define in more detail the quantitative aspects the recicling features. We are disinclined to believe that the protein residues can be so cleanly removed.

In general, the logical fluency of the work should be improved, since now the results are present as a pile of dry paragraphs, without a structure -activity red line.

These suggestions are appreciated formally as minor revision, but otherwise we expect for rather extensive improvement of the text. 

Author Response

Dear Editor,

On behalf of the authors, I thank the Reviewers for the comments, questions and remarks. We tried to take into account the comments and wishes of the reviewers. Corrections made to the text of the manuscript are shown in color.

Below are our responses to reviewer questions in order

Reviewer 2

  1. Please define, in causal manner, how the magnetic properties are working for the declared application purposes.

                        ­ The magnetic separation based on starch-MNPs eliminates centrifugation, filtration and the use of expensive chromatographic systems and columns for the target protein purification.

  1. Also, make explicit the implied recombination mechanism. What’s about specificity?

                        ­ It is not quite clear what the reviewer means by recombination mechanism.

The target proteins have been fused with maltose-binding protein (MBP). MBP is monomeric globular protein with molecular weights 42 kDa, containing a single ligand-binding site for α-(1–4) maltodextrin (Kd ∼3.5 μM). There is a specific interaction between the maltose-binding protein and amylose, which is part of the starch, on the surface of the starch-MNPs. When crude lysate of bacterial cells incubated with starch-MNPs the only protein molecules, containing maltose-binding protein were bound by nanoparticles. Unbound proteins washed away by buffer and after that specifically bound target protein eluted by solution of 10 mM maltose, which competes with amylose for binding sites of MBP. The specificity of the purification process is ensured by the natural affinity of MBP for α-(1–4) maltodextrin and can be examined by SDS-electrophoresis data. Some explanations have been added to the manuscript to make it more understandable.

 3. Add naked MNPs reference to ftir spectrum.

                        ­ Unfortunately, we do not have the opportunity to add these measurements, but we have significantly expanded the discussion of infrared spectroscopy results.

  1. To define the Fe-O bonding some x-ray surface spectroscopy would be useful.

                        Unfortunately, we do not have the opportunity to add these measurements, however, there are FTIR and Mössbauer spectroscopy data on the structure of the magnetic particle and the nature of the Fe-O bonds.

  1. Explain how the dicrimination of bonded starch layer from supllemenary outer ones is possible.

                        The binding of starch to the nanoparticle was determined by the change in absorption peaks. We have described in more detail in the FTIR discussion.

  1. How tick is the starch coating? Can it be stripped?

                        We did not accurately determine the thickness of the starch. According to our estimates, the thickness is several tens of nanometers.

We have shown the repeated use of the obtained particles for protein purification, as well as long-term storage (at least 6 months) without loss of properties. This confirms the stability of the immobilization of the cornstarch layer.

  1. Define in more detail the quantitative aspects the recicling features. We are disinclined to believe that the protein residues can be so cleanly removed.

                      ­ It is the common practice to reuse of chromatography sorbent after washing with elution buffer followed by equilibrated with buffer for binding.The experiment on nonspecific irreversible adsorption of protein on starch-MNPs have been carried out and described in the main text of manuscript. The total amount of protein taken for chromatography and the total amount of proteins in all eluted fractions was measured spectrophotometrically by Lowry. The amount of protein before and after chromatography was practically the same (3.94 mg of proteins of lysate taken for chromatography and 3.86 mg is the amount of proteins summed up from all eluted fractions). This indicates no protein loss associated with nonspecific irreversible adsorption on starch-MNPs. The data have been added to the manuscript.

  1. In general, the logical fluency of the work should be improved, since now the results are present as a pile of dry paragraphs, without a structure -activity red line.

                        ­ The mns text has been edited.

The authors hope that the changes made have significantly improved the manuscript and made it more understandable and useful for the reader.

On behalf of coauthors,

 Dr. Ludmila A. Frank.

Round 2

Reviewer 1 Report

The paper is very interesting and it is well written
In my opinion, you can be accepted this version.